# Replication of the *Salmonella* Genomic Island 1 (SGI1) triggered by helper IncC conjugative plasmids promotes incompatibility and plasmid loss

**Kévin T. Huguet** [ID], **Nicolas Rivard** [ID], **Daniel Garneau** [ID], **Jason Palanee** [ID], **Vincent Burrus** [ID] *

Département de biologie, Université de Sherbrooke, Sherbrooke, Québec, Canada

* vincent.burrus@usherbrooke.ca

**Data Availability Statement:** All relevant data are within the manuscript and its Supporting Information files.

## Abstract

The mobilizable resistance island *Salmonella* genomic island 1 (SGI1) is specifically mobilized by IncA and IncC conjugative plasmids. SGI1, its variants and IncC plasmids propagate multidrug resistance in pathogenic enterobacteria such as *Salmonella enterica* serovars and *Proteus mirabilis*. SGI1 modifies and uses the conjugation apparatus encoded by the helper IncC plasmid, thus enhancing its own propagation. Remarkably, although SGI1 needs a coresident IncC plasmid to excise from the chromosome and transfer to a new host, these elements have been reported to be incompatible. Here, the stability of SGI1 and its helper IncC plasmid, each expressing a different fluorescent reporter protein, was monitored using fluorescence-activated cell sorting (FACS). Without selective pressure, 95% of the cells segregated into two subpopulations containing either SGI1 or the helper plasmid. Furthermore, FACS analysis revealed a high level of SGI1-specific fluorescence in IncC⁺ cells, suggesting that SGI1 undergoes active replication in the presence of the helper plasmid. SGI1 replication was confirmed by quantitative PCR assays, and extraction and restriction of its plasmid form. Deletion of genes involved in SGI1 excision from the chromosome allowed a stable coexistence of SGI1 with its helper plasmid without selective pressure. In addition, deletion of *S003* (*rep*) or of a downstream putative iteron-based origin of replication, while allowing SGI1 excision, abolished its replication, alleviated the incompatibility with the helper plasmid and enabled its cotransfer to a new host. Like SGI1 excision functions, *rep* expression was found to be controlled by AcaCD, the master activator of IncC plasmid transfer. Transient SGI1 replication seems to be a key feature of the life cycle of this family of genomic islands. Sequence database analysis revealed that SGI1 variants encode either a replication initiator protein with a RepA_C domain, or an alternative replication protein with N-terminal replicase and primase C terminal 1 domains.

**Funding:** This work was supported by a Discovery Grant (2016-04365) from the Natural Sciences and Engineering Research Council of Canada (NSERC, https://www.nserc-crsng.gc.ca/index_eng.asp) and a Project Grant (PJT-153071) from the Canadian Institutes of Health Research (CIHR, https://cihr-irsc.gc.ca/e/193.html) to V.B. K.T.H. was supported by a postdoctoral fellowship (SPE20170336797) from the Fondation de la Recherche Médicale (FRM, France, https://www.frm.org/). N.R. is the recipient of an Alexander Graham Bell Canada Graduate Scholarship from the NSERC. The funders had no role in study design, data collection and analysis, decision to publish, or preparation of the manuscript.

**Competing interests:** The authors have declared that no competing interests exist.

## Author summary

The *Salmonella* genomic island 1 (SGI1) and its variants propagate multidrug resistance in several species of human and animal pathogens with the help of IncA and IncC conjugative plasmids that are absolutely required for SGI1 dissemination. These helper plasmids are known to trigger the excision of SGI1 from the chromosome. Here, we found that IncC plasmids also trigger the replication of the excised, circular form of SGI1 by enabling the expression of an SGI1-borne replication initiator gene. In return, high-copy replication of SGI1 interferes with the persistence of the IncC plasmid and prevents its cotransfer into a recipient cell, thereby allowing integration and stabilization of SGI1 into the chromosome of the new host. This finding is important to better understand the complex interactions between SGI1-like elements and their helper plasmids that lead to widespread and highly efficient propagation of multidrug resistance genes to a broad range of human and animal pathogens.

## Introduction

In the vast world of mobile genetic elements, mobilizable genomic islands are increasingly recognized as key players in the global spread of multidrug resistance. To date, more than 30 islands mobilized in *trans* by helper conjugative plasmids have been described, most carrying drug or heavy metal resistance genes [1]. One of the most atypical mobilizable genomic islands currently known is the 43-kb *Salmonella* Genomic Island 1 (SGI1) that was first described in 2000 in the multidrug-resistant epidemic strain of *Salmonella enterica* serovar Typhimurium DT104 [2,3]. SGI1 is integrated in the 3' end of *trmE*, a gene coding for a GTPase involved in the modification of U34 in tRNA, and carries at its 3' end a complex class 1 integron named In104, which confers resistance to β-lactams, chloramphenicol, tetracycline, streptomycin, and sulfamethoxazole (Fig 1A) [3]. The discovery of dozens of SGI1 variants with a large variety of resistance profiles revealed the high plasticity of the class 1 integron [4]. SGI1 and its variants have been described in a wide range of *S. enterica* serovars, in *Proteus mirabilis*, *Morganella morganii*, *Acinetobacter baumanii*, and more recently in avian pathogenic *Escherichia coli* [5–9]. The identification of SGI1 and variants in a broad range of species is suggestive of the high diffusion capacity of this mobile element.

Although SGI1 is not self-transmissible, it is specifically mobilized by conjugative plasmids of the closely related IncA and IncC incompatibility groups [10,11]. IncC plasmids are widespread in several pathogenic species of *Enterobacteriaceae* isolated from food products, food-producing animals and humans. IncC plasmids have been sporadically found in African seventh-pandemic isolates of *Vibrio cholerae* O1 El Tor, the causative agent of the diarrheal disease cholera [12,13]. Nowadays IncC plasmids are globally distributed and considered to be important contributors to the diffusion of drug resistance genes, including New Delhi metallo-β-lactamase genes ($bla_{NDM}$) that confer resistance against most β-lactams including carbapenems [14,15]. Albeit not as epidemiologically successful, a highly conjugative IncA plasmid was recently found to participate in the emergence of carbapenemase-producing *Enterobacteriaceae* ($bla_{VIM-1}$) that colonized the gut of patients in a university hospital in Italy [16].

Several genes carried by SGI1 have been shown to be essential for its transfer. For instance, *int* and *xis* code for the integrase and recombination directionality factor that catalyze the integration of SGI1 into the 3' end of *trmE* and its excision from the chromosome (Fig 1A) [10]. Furthermore, the recently identified *mpsA* and *mpsB* code for key mobilization factors that recognize the upstream *cis*-acting origin of transfer loci (*oriT*) [17]. Nevertheless, like all

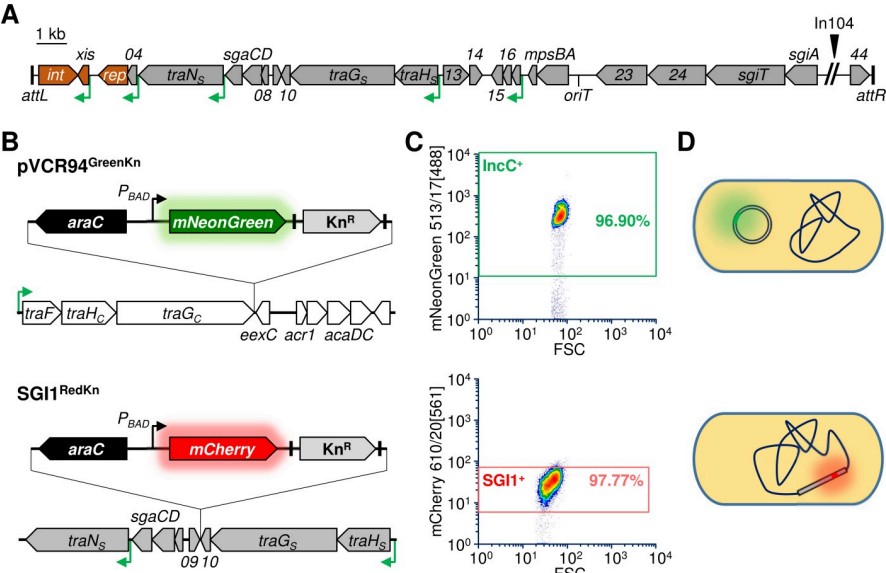

**Fig 1. Monitoring of pVCR94 and SGI1 in cells by flow cytometry (FC). A.** Schematic representation of chromosomally integrated SGI1. ORFs are represented by gray arrows. Gene numbers correspond to the last digits of locus tags S0xx in the Genbank accession number AF261825. ORFs of interest in this work are shown in brown. AcaCD binding sites are depicted by green angled arrows. Left (*attL*) and right (*attR*) junctions with the chromosome are indicated by black bars at the ends. The position of the complex class 1 integron In104 (multidrug resistance region) is indicated by a black arrowhead. **B.** Fluorescent reporter genes *mNeonGreen* (green arrow) and *mCherry* (red arrow) under the control of the $P_{BAD}$ promoter were inserted in pVCR94 between *traG_C* and *eexC*, and in SGI1 between *S009* and *S010*. **C.** Representative flow cytometric scatter plots of green and red fluorescence of *E. coli* KH95 bearing either pVCR94$^{GreenKn}$ or SGI1$^{RedKn}$. **D.** Schematic representation of the labelled elements in the cells. SGI1$^{RedKn}$ is integrated into the chromosome at the 3' end of *trmE*.

mobilizable genomic islands, SGI1 lacks a complete set of genes coding for a fully functional type IV secretion system (T4SS) to be self-transmissible and relies specifically on the T4SS encoded by IncA or IncC helper plasmids to transfer to a new host [11]. However, unlike any other known mobilizable genomic island, SGI1 alters the mating apparatus encoded by IncC plasmids by replacing the T4SS proteins TraN_C, TraG_C, and TraH_C with its own distant orthologues TraN_S, TraG_S, and TraH_S, respectively [18]. This remodeling not only seems to enhance SGI1 transfer at the expense of the helper plasmids but also disables entry exclusion mediated by IncA and IncC plasmids, thereby enabling the transmission of SGI1 to bacterial cells that bear such plasmids [18,19]. Expression of *xis*, *traN_S* and *traHG_S* of SGI1 has been shown to be activated by the master activator complex AcaCD encoded by IncA and IncC plasmids [18,20,21]. Therefore, excision of SGI1 from the chromosome occurs only in the presence of a helper plasmid. Together with the presence of the *sgiAT* toxin-antitoxin system, tight regulation of excision contributes to the high stability of SGI1 in the absence of an IncC plasmid [22,23].

SGI1 and IncC plasmids are never found together in natural isolates [24,25]. In fact, SGI1 was found to destabilize IncA and IncC plasmids after a few generations [23,26]. Conversely, the presence of an IncC plasmid has been shown to enhance the recombination rate within SGI1, leading to the generation of SGI1 deletion variants [22]. Conjugation-dependent rolling-circle replication of SGI1 has been suggested to be a possible explanation for this instability. Here, we characterize the cause of the apparent incompatibility between SGI1 and IncC plasmids using flow cytometry to monitor the segregation of the genomic island and its helper plasmid expressing different fluorescent proteins. While confirming the incompatibility, we

observed an unexpected high-copy replication of the excised form of SGI1 that only occurred in the presence of the helper plasmid. Replication was confirmed by real-time quantitative PCR as well as DNA extraction and restriction of the replicative form. We identified an iteron-based origin of replication (*oriV*) downstream of a replication initiator gene whose expression is activated by an AcaCD-dependent promoter. Further investigations showed a clear link between incompatibility and SGI1 replication as replication-deficient mutants allowed stable cohabitation with the helper plasmid, even in the absence of any selection pressure.

## Results

### Monitoring of SGI1 and pVCR94 incompatibility by flow cytometry

To monitor the fate of SGI1 and its helper IncC plasmid in cell populations by flow cytometry, we constructed SGI1$^{RedKn}$ and pVCR94$^{GreenKn}$ by inserting genes expressing red or green fluorescent reporter proteins between two converging genes to avoid interference with key functions (Fig 1B). Flow cytometry monitoring of *E. coli* cells bearing either pVCR94$^{GreenKn}$ or SGI1$^{RedKn}$ revealed that ~97% and ~98% of the cells expressed the green and red fluorescent reporters under inducing conditions, respectively (Fig 1C). These values were the maxima that could be reached for pure populations bearing either element. In this context, we used the green and red fluorescence as a proxy for the presence of the IncC plasmid and SGI1 in cells and assumed that pVCR94$^{GreenKn}$ was present in a single copy per cell, and that a single copy of SGI1$^{RedKn}$ was inserted at *trmE* per chromosome (Fig 1D).

To measure by flow cytometry the previously reported incompatibility between SGI1 and IncC plasmids [23,26], SGI1$^{RedKn}$ and kanamycin-sensitive pVCR94$^{Green}$ were introduced together by conjugation in *E. coli*, and the fluorescence of the resulting cell population was monitored over a 3-day period (~54 generations) in the absence of antibiotics (Fig 2A). While at the beginning of the assays (G0), 97.5% of the cells exhibited both green and red fluorescence, only 51% of the cells produced both signals after 18 generations (G18), and less than 0.5% after 54 generations (G54), thereby suggesting that SGI1 and pVCR94 are incompatible. To confirm that our observation resulted from plasmid or SGI1 loss, not from inactivating mutations in the reporter genes, antibiotics kanamycin and chloramphenicol were added at day 3. Selective pressure for both elements restored dual fluorescence in more than 97% of the cells, and the population of element-free cells nearly vanished (Fig 2A, ATB).

Analysis of the individual scatter plots for the red and green channels revealed additional information regarding the incompatibility phenomenon. While the gradual loss of pVCR94$^{Green}$ over time yielded two distinct cell populations (IncC$^+$ and IncC$^-$ cells) in the green channel, three distinct cell populations emerged in the red channel (Fig 2C, bottom). At G18, ~57% of the cells exhibited high-intensity red fluorescence like 95% of the cells at G0. A second population emerged with a red fluorescent signal weaker than the initial population, but comparable to the intensity emitted by IncC plasmid-free cells bearing a single copy of SGI1$^{RedKn}$ integrated at *trmE* (Fig 1C, right). The presence of two cell subpopulations suggests a differentiation into two types of SGI1$^+$ cells, one bearing a single integrated copy of SGI1$^{RedKn}$ and the other bearing multiple copies. At G36 and G54, the subpopulation of cells with high-intensity red fluorescence collapsed, while the subpopulations with low or no red signal increased. The collapse of the high-intensity red fluorescent signal subpopulation correlated with the loss of green signal, i.e. the loss of pVCR94$^{Green}$ (Fig 2C, top), indicating that high-intensity red fluorescence was dependent on the presence of the IncC plasmid. Based on these observations, we hypothesized that in IncC$^+$ cells, not only does SGI1 excise from the chromosome, but also undergoes a transient replication cycle that increases the production of mCherry protein in cells.

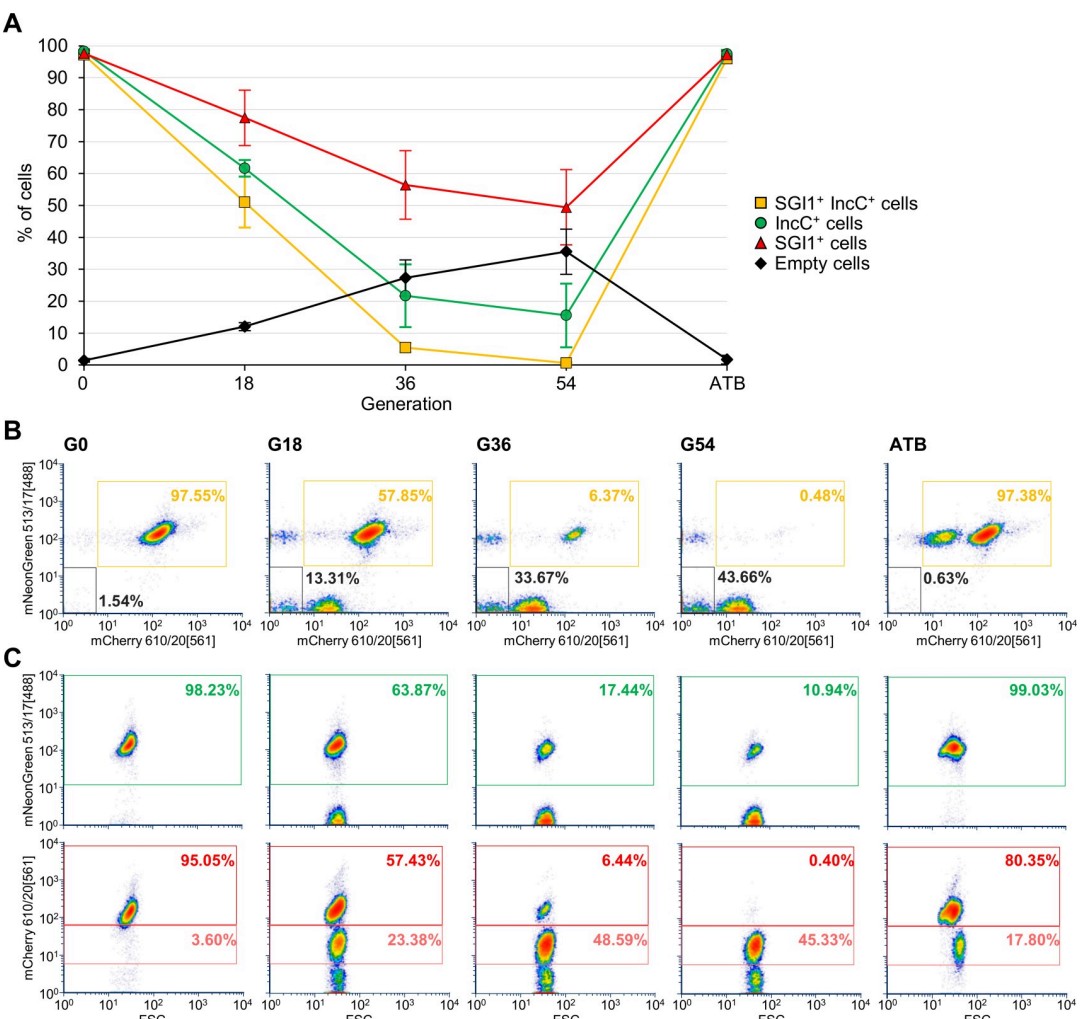

**Fig 2. Incompatibility between SGI1 and IncC plasmids is trackable by FC. A.** Evolution of the percentage of *E. coli* KH95 cells bearing SGI1$^{RedKn}$ (SGI1) and pVCR94$^{Green}$ (IncC) and grown over 54 generations in the absence of antibiotics as monitored using FC. Plots represent the mean and standard error of the mean obtained from three independent experiments. ATB is a recovery control culture at G54 in LB with selective pressure for both elements. **B.** Representative FC density plots of the data presented in panel A mapping the green signal (513 nm) as a function of the red signal (610 nm). **C.** Representative FC density plots of fluorescence intensity over forward scatter corresponding to data presented in panel B. Color keys are identical in all panels. In panel C, pink and red indicate cells producing low- and high-intensity red fluorescence, respectively.

## Active replication of SGI1

To test whether SGI1 is able to replicate, we first prevented its excision from the chromosome by using a Δ*int* mutant of SGI1 that is unable to excise and remains permanently integrated in the chromosome [10]. pVCR94$^{Green}$ was then introduced into *E. coli* bearing SGI1$^{Red}$ Δ*int* locked in *trmE*. We did not observe cells producing a high level of red fluorescence (Fig 3A, compare WT and Δ*int*, and S1A Fig), suggesting that SGI1 locked into the chromosome is unable to replicate, even in IncC$^{+}$ cells.

To confirm SGI1 replication, we attempted to lock SGI1$^{Red}$ Δ*int* out of the chromosome in a circular, replicative form that we called pSGI1. To do this, *int* was deleted while SGI1$^{Red}$ was in its excised, circular state in cells that contained spectinomycin-resistant pVCR94$^{GreenSp}$ with antibiotic selection to counter incompatibility. This can be achieved because the IncC

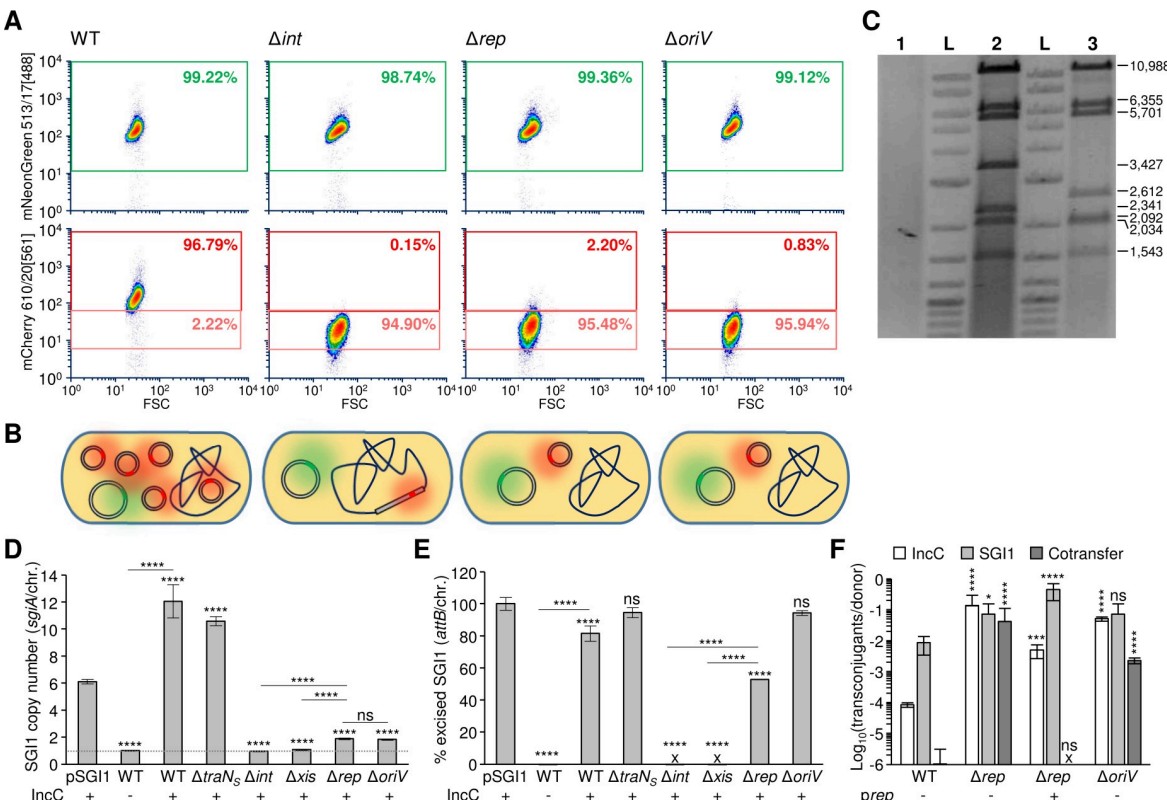

**Fig 3. Replication of SGI1 in the presence of an IncC plasmid. A.** Representative FC density plots of green and red fluorescence at G0 of cells bearing pVCR94$^{Green}$ (IncC) and SGI1$^{Red}$ (WT), its locked-in Δ*int* mutant and its Δ*rep* and Δ*oriV* mutants. **B.** Schematic representation of the cell content in the different mutants of panel A. **C.** Ethidium bromide-stained 0.8% agarose gel electrophoresis of EcoRI-digested plasmid DNA preparation of the same IncC$^+$ cells bearing SGI1$^{Red}$, its locked-in Δ*int* mutant or locked-out Δ*int* mutant (pSGI1). Lane 1, locked-in *trmE*::(SGI1$^{Red}$ Δ*int*::*aph*); Lane 2, SGI1$^{Red}$; Lane 3, pSGI1; Lane L, 2-Log DNA ladder. **D.** Effect of deletions on SGI1 copy number. Quantification by qPCR of the SGI1 copy number as measured by the *sgiA*/chromosome ratios at G0. **E.** Effect of deletions on SGI1 excision. Quantification by qPCR of the percentage of cells at G0 that contain excised SGI1 as a measure of unoccupied *attB* sites per 100 chromosomes. "x" indicates that excision was below the limit of detection (<10$^{-5}$%). Assays were carried out in *E. coli* KH95 carrying (+) or lacking (-) the helper IncC plasmid pVCR94$^{Green}$ or pVCR94$^{GreenSp}$ (for pSGI1 only). The bars represent the mean and standard error of the mean obtained from at least three independent experiments. Statistical analyses were carried out on the values using the one-way ANOVA with Tukey's multiple comparison test. **F.** Effect of SGI1 replication on conjugative transfer of SGI1 and its helper plasmid. Conjugation assays were carried out using *E. coli* KH95 as the donor and *E. coli* VB113 as the recipient. When indicated (+), Rep was provided in *trans* from p*rep*. Transfer frequencies are expressed as the number of transconjugant per donor CFUs. Statistical analyses were carried out on the logarithm of the values using the one-way ANOVA with Sidak's post-test to compare each mutant set relatively to SGI1$^{Red}$ (WT) control. The bars represent the mean and standard deviation values obtained from at least three independent experiments. Statistical significance is indicated as follow: ****, $P<0.0001$; ***, $P<0.001$; *, $P<0.05$; ns, not significant.

plasmid stimulates SGI1 excision by activating the expression of *xis* through AcaCD [20]. Plasmid DNA was extracted from cells carrying the resulting mutant and digested with EcoRI. A clear restriction pattern was expected only in conditions that would allow replication of a high number of SGI1 molecules. As expected, no restriction pattern was visible for the strain bearing chromosomally locked-in SGI1$^{Red}$ Δ*int*, (Fig 3C, lane 1). In contrast, EcoRI restriction patterns were easily detected for SGI1$^{Red}$ or locked-out pSGI1 (Fig 3C, lanes 2 and 3). Only high-copy number replication of SGI1 can explain our ability to extract, recover, and visualize plasmid DNA of the excised form of this genomic island.

To measure the variation of SGI1 copy number due to replication, we quantified SGI1 copy number per chromosome using real-time quantitative PCR (qPCR). In the IncC$^+$ strain, SGI1$^{Red}$ copy number increased 12-fold compared to the IncC$^-$ strain and pSGI1 was

maintained at ~6 copies per cell (Fig 3D). Altogether, these results confirm that the subpopulation of cells that produced high-intensity red fluorescence in IncC$^+$ cells corresponds to cells containing an excised replicative form of SGI1.

## *rep* is required for SGI1 replication

Although its role in replication has never been demonstrated, the putative protein encoded by *S003*, also known as *rep*, contains a RepA_C domain (Pfam PF04796). The predicted translation product of *rep* shares 60% identity (78% similarity) with the putative replication initiator protein RepA encoded by plasmid pXAC64 of *Xanthomonas axonopodis* pv. *citri* [27]. To determine whether *rep* plays a role in SGI1 replication, we constructed a *rep* deletion mutant of SGI1$^{Red}$, and tested its ability to replicate in cells bearing pVCR94$^{Green}$ using flow cytometry and qPCR to quantify its rate of excision and copy number. The data were compared with wild-type SGI1$^{Red}$ and its Δ*traN$_S$* mutant as positive controls, and its Δ*int* and Δ*xis* mutants as negative controls. Excision of wild-type SGI1$^{Red}$ was barely detectable in IncC$^-$ cells (0.0000838%) whereas 81% of the cells contained excised SGI1$^{Red}$ in IncC$^+$ cells (~7-log increase) (Fig 3E). SGI1$^{Red}$ and its Δ*traN$_S$* mutant behaved alike in IncC$^+$ cells, with virtually all cells bearing the excised, replicative form (>80% excision and >10 copies per chromosome) (Fig 3D and 3E). As expected, deletion of *int* or *xis* abolished excision, and led to a low copy number of SGI1$^{Red}$ that was comparable to the wild-type in IncC$^-$ cells. In contrast, while the Δ*rep* mutant retained the ability to excise from the chromosome (>52% excision), it remained under 2 copies per cell (Fig 3D and 3E), thereby supporting a key role of *rep* in SGI1 replication.

Flow cytometric data confirmed the inability of the Δ*int*, Δ*xis* and Δ*rep* mutants to replicate. Only low-intensity red fluorescence was detected despite the presence of the helper IncC plasmid, unlike the Δ*traN$_S$* mutant (Fig 3A, and S1A, S2, S3A and S4A Figs). Complementation of Δ*int*, Δ*xis* and Δ*rep* mutations by expressing the corresponding gene under the control of the $P_{BAD}$ promoter restored the high-intensity red fluorescence phenotype of the cells, indicating a restoration of SGI1 replication (S1B, S3B and S4B Figs).

## Identification of the origin of replication (*oriV*) of SGI1

Based on the presence of a *rep* gene and the inability of SGI1 Δ*rep* to replicate, we searched for nearby intergenic regions that could act as a potential *oriV* locus. Sequence analyses revealed a putative *oriV* immediately downstream of *rep* (Fig 4A). This locus contains four GC-rich 17-bp direct repeats (iterons) and a putative DnaA box flanked by two AT-rich regions. The longest AT-rich region contains three direct and inverted copies of an imperfect 8-bp repeat (Fig 4B). This configuration resembles the origin structure of typical iteron-containing replicons [28]. To confirm that this putative *oriV* is essential for SGI1 replication, it was replaced with a Kn$^R$ cassette without any alteration of the promoter region of *xis*. The resulting 255-bp deletion abolished SGI1$^{Red}$ replication as shown by the lack of highly red fluorescent cells despite the presence of pVCR94$^{Green}$ (Fig 3A). Moreover, SGI1$^{Red}$ Δ*oriV* retained the ability to excise at the same level as wild-type SGI1$^{Red}$; however, its copy number per cell was considerably reduced and remained comparable to SGI1$^{Red}$ in IncC$^-$ cells or to its Δ*rep* mutant in IncC$^+$ cells (Fig 3D and 3E). Together, these results indicate that the sequence downstream of *rep* contains *oriV*.

## Expression of SGI1 *rep* is activated by IncC plasmid encoded AcaCD

To test the role of the IncC helper plasmid in SGI1 replication, the locked-out replicative form pSGI1 was subjected to the cohabitation test with pVCR94$^{GreenSp}$ for 54 generations in the

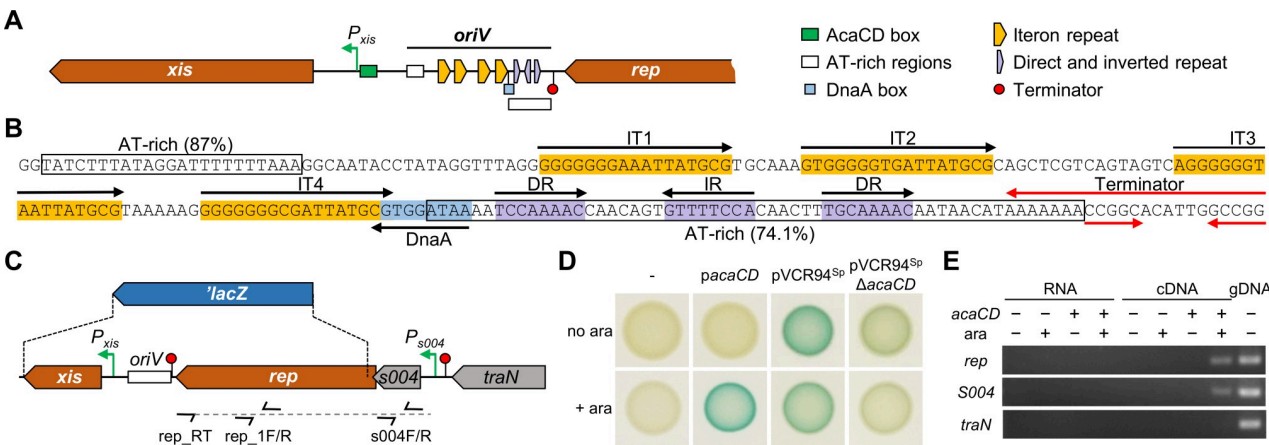

**Fig 4. Identification of the *oriV* locus of SGI1. A.** Schematic map of the *xis-S003* intergenic region of SGI1 containing *oriV*. **B.** Sequence of the *oriV* region of SGI1. Sequence features are indicated and color-coded as in panel A. The stem-loop of the putative rho-independent transcriptional terminator located downstream of *S003* has a calculated free energy ($\Delta G$) of -8.9 kcal/mol (ARNold). **C**. Schematic representation of the *S004-rep* locus and translational *rep'-'lacZ* fusion. The translational *lacZ* fusion was introduced at position 3,246 after the fifth codon of *rep* (refer to S6A Fig for details). The relative positions of reverse transcription primer rep_RT as well as PCR primers used to amplify *rep* and *S004* fragments are indicated. The dotted line shows reverse transcription product. **D**. *rep* expression is controlled by AcaCD. β-galactosidase assays of the translational *rep'-'lacZ* fusion in SGI1 Δ*xis* performed in IncC-free cells (-), and in the presence of pVCR94 or p*acaCD* without (no ara) or with arabinose (+ ara). **E**. Analysis on a 1.5% agarose gel from an assay to amplify *rep* and *S004* on the rep_RT cDNA. Genomic DNA (gDNA) and RNA samples in the absence of reverse transcriptase (noRT) were used, respectively, as positive and negative PCR controls.

absence of selective pressure and analyzed by flow cytometry (Fig 5A and S5 Fig). As observed for SGI1^Red, incompatibility led to the loss of both elements. However, while pVCR94^GreenSp

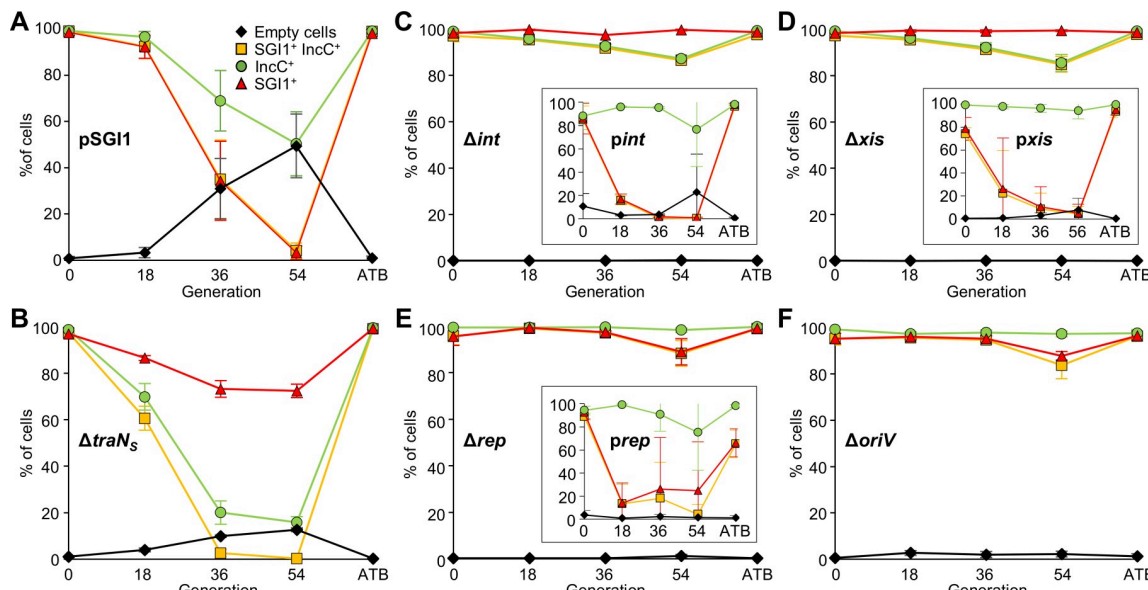

**Fig 5. Impact of SGI1 replication on incompatibility.** Evolution of the percentage of *E. coli* KH95 cells bearing either pVCR94^GreenSp (IncC) and pSGI1 (A) or pVCR94^Green (IncC) and the Δ*traN_S* (B), Δ*int* (C), Δ*xis* (D), Δ*rep* (E), or Δ*oriV* (F) mutants of SGI1^Red (SGI1) over 54 generations in the absence of antibiotics as monitored by FC. Inserts for panels C, D and E show complementation experiments using the designated expression plasmid and pVCR94^GreenSp. Conditions were identical to Fig 2A. Plots represent the mean and standard error of the mean obtained from three independent experiments. Representative density plots of mNeongreen intensity over mCherry intensity and their corresponding plots of fluorescence intensity over forward scatter are shown for each condition in S1–S5 and S7 Figs.

persisted in 50% of the cells, pSGI1 was lost in virtually all cells after 54 generations. The other half of the cell population consisted of cells lacking both elements. This behavior is in stark contrast with SGI1$^{Red}$ which was retained chromosomally integrated after 54 generations in more than 45% of the cells while the IncC plasmid was retained in less than 11% of the cells, with less than 0.5% of the cell population containing both elements (Fig 2). Strict correlation of the plots of SGI1$^+$ cells and SGI1$^+$ IncC$^+$ cells suggests that pSGI1 replication and stability are directly linked to the IncC plasmid. As pSGI1 did not integrate into the chromosome due to the missing *int* gene, we hypothesize that the IncC plasmid codes for an essential factor that sustains pSGI1 replication.

Our previous identification of an AcaCD binding site upstream of the *S004-rep* gene cluster suggests that *rep* could be expressed under the control of the IncC plasmid-encoded transcriptional activator AcaCD (Fig 1) [20]. To test this hypothesis, we first constructed a translational fusion between the fifth codon of *rep* and the eighth codon of *lacZ* in SGI1$^{Kn}$ (Fig 4C and S6A Fig). This fusion also removes *xis* and *rep*, thereby preventing SGI1 excision and replication. β-galactosidase activity of the *rep'-'lacZ* fusion was then used as a proxy to measure the level of *rep* expression (transcription and translation) in the presence or absence of AcaCD. β-galactosidase activity was undetectable in IncC$^-$ cells or in cells lacking p*acaCD* (Fig 4D). In contrast, when AcaCD was provided by p*acaCD* with arabinose induction, we observed a 172±21-fold increase in β-galactosidase activity, and the presence of pVCR94$^{Sp}$ resulted in a 11.9±0.3-fold increase compared to strain containing its Δ*acaCD* mutant. Comparable results were observed using a fusion that retained *xis*, allowing SGI1 excision but not its replication (S6B and S6C Fig).

Furthermore, to confirm that *S004* and *rep* are part of the same mRNA transcript initiated at P$_{S004}$, we conducted a PCR assay aimed at amplifying both genes from the same cDNA. To do this, RNA was extracted from cells bearing SGI1$^{Red}$ alone, or in the presence of p*acaCD* with or without arabinose. *rep*-specific mRNA was reverse transcribed using a primer located at the 3' end of *rep* and used in a PCR assay to amplify fragments located within *S004* and *rep* (Fig 4C). Our results confirmed that these two genes are cotranscribed and show unambiguously that *S004* and *rep* are part of the same operon that is activated by AcaCD (Fig 4E).

## SGI1 replication promotes instability of the IncC plasmid and inhibits cotransfer

We observed that SGI1 replication and IncC instability are strongly correlated. SGI1 replication resulted in a strong incompatibility phenotype as shown by low counts of SGI1$^+$ IncC$^+$ cells that remained at G36 and G54 (Figs 2A and 5B and S2 Fig). In contrast, mutants impaired for excision (Δ*int* and Δ*xis*) or replication (Δ*rep* and Δ*oriV*) exhibited a drastically reduced incompatibility as over 85% of the cells retained both SGI1 and the IncC plasmid at G54 (Fig 5C–5F). Therefore, replication of SGI1 is responsible for the incompatibility phenotype observed with IncC plasmids.

Furthermore, when excision of SGI1 was abolished, we observed that retention of SGI1 was favored over pVCR94 (Δ*int*, 99.1% vs 87.5%, Δ*xis*, 99.6% vs 87.3%, respectively) (Fig 5C and 5D, and S1A and S3A Figs). Conversely, retention of pVCR94 was favored when SGI1 was able to excise but unable to replicate (Δ*rep*, 97.6% vs 94.3%, Δ*oriV*, 92.6% vs 76.7%, respectively) (Fig 5E and 5F, and S4A and S7A Figs). These observations suggest that replication plays a key role in the stability of SGI1 and improves its retention in cell populations colonized by IncC plasmids.

SGI1 and several of its variants were previously shown to inhibit cotransfer of IncC plasmids such as pRMH760 and pVCR94 [20,26]. To test whether SGI1 replication could play a

role in this inhibition, we carried out mating assays using the Δ*rep* and Δ*oriV* mutants of SGI1. These mating assays revealed that the frequency of cotransfer was comparable to the transfer of each single element (Fig 3F), indicating that both mutations alleviated cotransfer inhibition. Furthermore, transfer of pVCR94 increased by 3 logs in both mutants suggesting that SGI1 replication strongly impairs the transfer of the IncC plasmids. We also observed that inhibition of cotransfer was restored as the frequency of cotransfer dropped below the detection limit when the Δ*rep* mutation was complemented by providing Rep from p*rep*. In this context, we observed a 27-fold reduction of pVCR94 transfer compared to the Δ*rep* mutant (Fig 3C), while transfer of SGI1 increased 6-fold, thereby suggesting that replication provides a significant advantage to SGI1 compared to the IncC plasmid.

## SGI1-like elements can encode an alternative replication initiator protein

A search for divergent SGI1 homologues sharing a genomic structure similar to SGI1 and GI-15 of *V. cholerae* O1 B33 (sulfamethoxazole and streptomycin resistance) revealed two SGI1-like elements with an alternative *rep* gene located at the same position as SGI1 *rep*. The first one, named GI*Vch*O27-1, was identified in the unique chromosome of *V. cholerae* O27 strain 10432–62, a strain isolated in the feces of a patient with diarrhea in 1962 in the Philippines [29]. The second one, named GI*Sen*-26, was found in the genome of *S. enterica* serovar Muenster 26, a multidrug resistant strain isolated from horse feces at a Texan equine referral hospital. GI*Sen*-26 carries a class I integron (*aadA1*, *sul1*, *qacEΔ1*), a Tn*3*-like transposon (*strAB*, *tetA*), and a mercury resistance transposon (*merRTPFADE*). Both GIs are integrated at *trmE* and encode a replication protein unrelated to SGI1 and GI-15 Rep (Fig 6). This alternative replication initiator protein contains an N-terminal replicase (PF03090) domain and an adjacent primase C terminal 1 (PriCT-1, PF08708) domain instead of the RepA_C domain. As expected

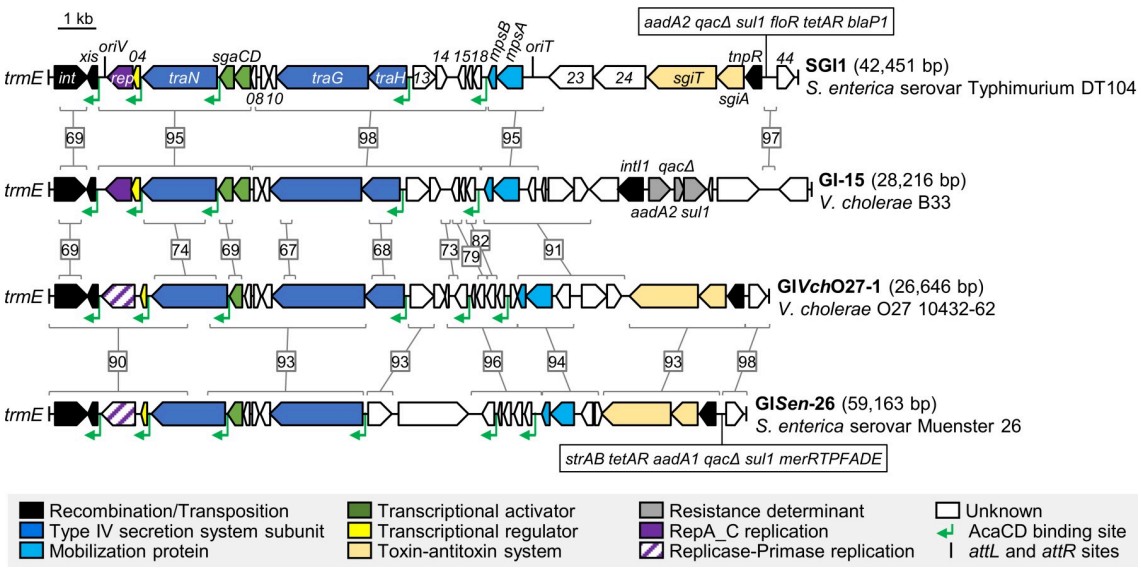

**Fig 6. Comparison of the genetic maps of 4 SGI1-like genomic islands.** Genomic islands are drawn to scale. The left and right junctions (*attL* and *attR*) within the host chromosome are indicated by black bars at the ends. ORFs with similar function are color coded as indicated in the figure. Green flags indicate the position and orientation of AcaCD binding sites predicted using MAST of the MEME suite [30]. Homologous regions are bracketed and linked by a gray line with the corresponding percentage of nucleotide identity. Genbank accession numbers: SGI1, AF261825.2; GI-15, AAWE01000022); GI*Vch*O27-1, CP010812; GI*Sen*-26, QDTO01000013.

in this context, SGI1 *oriV* is missing, and likely replaced by an alternative *oriV* that remains to be characterized.

Furthermore, although the alternative *rep* gene is located upstream of an *S004* homologue, the putative translation product of this gene shares only 35% with S004, suggesting a functional replacement of the *oriV-rep-S004* fragment. However, as described in SGI1 and GI-15, the *S004* homologue is preceded by a predicted AcaCD-dependent promoter. BlastP analysis using the protein sequence of GI*Sen*-26 RepA revealed identical proteins (100% identity) encoded by SGI1-like elements in *S. enterica* (serovars Heidelberg, Typhimurium, Agona, Senftenberg, Infantis, Alachua, Cerro, Saint-Paul and Montevideo), *Rheinheimera nanhaiensis*, *V. cholerae*, *Vibrio mimicus*, *Shewanella algae*, *Shewanella fodinae*, *Escherichia albertii*, *E. coli* and *Proteus mirabilis*, including in the multidrug resistance-conferring PGI1-PEL of *P. mirabilis* PEL isolated in urine from a patient hospitalized in France (*aacA4*, *aadB*, *dhfrA1*, *qacEΔ1*, *sulI*, *bla*$_{VEB-6}$, *aphA6*, *bla*$_{NDM-1}$, *bleMBL*, *bla*$_{DHA-1}$, *merRFPTADE*) [31]. Therefore, although alternative replicons can be found in SGI1-like elements, autonomous replication appears to be an essential feature of the life cycle of SGI1-like GIs.

## Discussion

Mobile genetic elements such as conjugative plasmids and mobilizable genomic islands are potent drivers of antibiotic resistance gene dissemination. A better understanding of their life-cycle and interactions is essential to devise novel strategies aimed at curbing their propagation. In this study, we unraveled a yet unforeseen aspect of the complex interactions between IncC plasmids and SGI1, two types of mobile genetic elements that are both frequently found in several species of *Enterobacteriaceae* including *Salmonella enterica*, *Morganellaceae* and *Vibrionaceae* [4,32–34]. Previous reports of incompatibility between IncC (and IncA) plasmids and SGI1 emphasize the complexity of these interactions, especially when considering that SGI1 needs IncC or IncA plasmids to propagate [20,23,26]. We undertook to investigate the underlying cause of incompatibility by labelling SGI1 and the helper IncC plasmid pVCR94 with fluorescent reporter genes to follow by flow cytometry their respective fates in a cell population. This strategy allowed us to confirm that SGI1 and IncC plasmids are incompatible as cells segregated into two subpopulations containing one or the other element after only a few generations. Furthermore, different cells contained different configurations of SGI1. Besides IncC plasmid-free cells that bear quiescent SGI1 integrated at *trmE*, we found that cells bearing an IncC plasmid contain SGI1 in an excised, high-copy replicative state. In IncC plasmid-free cells, virtually no excision of SGI1 was detected, whereas in IncC$^+$ cells, up to 12 copies of SGI1 could be found in more than 80% of the cells. DNA of the replicative form of SGI1 (pSGI1) was easily extracted and profiled by EcoRI restriction, confirming the high copy number per cell.

We found that replication requires prior excision of SGI1 from the chromosome which occurs only in the presence of an IncA or IncC plasmid [10,11]. Hence, replication of SGI1 mutants lacking *xis* or *int* could not be detected, even if initiation of replication of integrated SGI1 cannot be ruled out. Our analysis of SGI1 sequence revealed an iteron-based replicon with a *rep* gene coding for an uncharacterized replication initiator protein. *rep* is preceded by *S004*, a gene coding for a protein of unknown function that contains a predicted helix-turn-helix domain (HTH_17, Pfam PF12728). Murányi *et al.* [35] presented evidence supporting an expression of *S004* driven from the AcaCD-responsive promoter $P_{S004}$, while expression of the downstream gene *rep* (*S003*) would be very low and driven from a constitutive weak promoter located within *S004* regardless of AcaCD. Their analysis of *rep* expression relied on β-galactosidase assays with a transcriptional *lacZ* fusion that substituted *lacZ* for *rep* at the start codon in

a multi-copy reporter plasmid containing most of *S004* and its promoter region (position 3,261–3,639 of SGI1) seemingly displacing *S004* stop codon. In contrast, we devised a translational *rep*'-'*lacZ* fusion constructed in SGI1. In this more natural context, our results show that *S004-rep* is an operon driven from $P_{S004}$, directly under the control of AcaCD. Hence excision and replication of SGI1 are coordinated events controlled by the same transcriptional activator produced by IncA and IncC plasmids. Consistent with this observation, we found that replication of pSGI1, the locked-out form of SGI1 Δ*int*, cannot persist in the absence of the helper IncC plasmid. Furthermore, consistent with AcaCD-induced expression of *rep*, destabilization of the IncC plasmid triggered by pSGI1, ultimately resulted in the concomitant loss of pSGI1 (Fig 5A).

We also found that although *S004-rep* expression was off in IncC⁻ cells, the operon was weakly expressed in the presence of a Δ*acaCD* mutant of the helper plasmid (Fig 4D). This observation suggests that another IncC plasmid-encoded factor triggers *rep* expression, directly or indirectly, perhaps through the activation of *sgaCD* (aka *flhDC*$_{SGI1}$) expression. *sgaCD* is carried by SGI1 and encodes a close homologue of AcaCD (89% and 67% identity for the C and D subunits, respectively) [21,35,36]. SgaCD has been shown to activate AcaCD-dependent promoters of SGI1, though at a lower level than AcaCD [35]. While the exact role of SgaCD in SGI1 biology is currently unknown, a natural variant of SGI1 that lacks *sgaCD*, SGI1-K, has been shown to be unable to destabilize the IncC plasmid pRMH760 [26], suggesting a possible link between SgaCD, SGI1 replication and IncC plasmid destabilization. Identification of the IncC plasmid-encoded factor and pathway that control *rep* expression in the absence of *acaCD* is ongoing.

Little is known about SGI1's replicon as the product of *rep* has not been characterized. To the best of our knowledge, the closest relative of SGI1 Rep is RepA of plasmid pXAC64 of *X. axonopodis* pv. *citri* that was identified by sequencing only and not studied further [27]. Therefore, additional studies are needed to better understand the molecular mechanism by which SGI1 replicates, the control mechanisms that affect the copy number of its excised form as well as the mechanism, whether direct or indirect, by which it affects the stability of IncC plasmids. We showed that suppression of SGI1 excision and/or replication functions (*xis*, *int*, *rep*, *oriV*) enables stable coexistence of SGI1 and its helper plasmid. Surprisingly, overexpression of *rep* in complementation assays led to a destabilization of SGI1 over the IncC plasmid (Fig 5E). High intracellular Rep concentration could inhibit replication through "handcuffing", a mechanism that couples replication origins via iteron-bound Rep proteins, turning off origin function as reported for the π replicase of plasmid R6K [37]. Replication of iteron-based plasmids involves the recruitment of the chromosomal replication initiator protein DnaA. We identified in the *oriV* of SGI1 a putative DnaA box and four 17-bp iterons (consensus RKGGGGGHRATTATGCG) (Fig 4A). IncA and IncC replicons also contain a single DnaA box and 11 to 14 copies of 19-bp iterons that differ in sequence (yaTRTGGGDNHgcTGCACG and yaTRTGGGNNcgcTGCACG, respectively) with SGI1 iterons [38,39]. Destabilization of IncA and IncC plasmids by SGI1 replication could result from the titration of host-encoded replication proteins by the *oriV* of replicating SGI1. DnaA titration by the DnaA box of multi-copy SGI1 could be a mechanism preventing proper initiation of IncA and IncC plasmid replication. Alternatively, SGI1 could produce a protein that interferes with key maintenance functions that are conserved in IncA and IncC plasmids such as replication (*repA*), partition (*parAB*, *053*) or post-segregational killing (*tad-ata*) [20,40]. This factor would be produced when SGI1 is excised and replicating, and not produced or produced in insufficient quantity when in single copy in the cell, that is in cells devoid of helper plasmid.

Integrative and conjugative elements (ICEs), such as ICE*Bs1* of *Bacillus subtilis*, ICE*St3* of *Streptococcus thermophilus*, Tn*916* of *Enterococcus faecalis*, R391 of *Providencia rettgeri* or

ICE*clc* of *Pseudomonas putida*, were shown to undergo transient replication after excision from the chromosome [41–45]. For these mobile elements, increased copy number results from an intercellular rolling-circle replication mechanism initiated at the *oriT* locus by the conjugative relaxase that is used as a replication initiator protein. In contrast, we found that replication of SGI1 relies on a dedicated *rep* gene and *oriV* that are distinct from the *oriT* and *mpsAB* mobilization genes [17]. In this respect, SGI1 resembles integrative and conjugative elements found in Actinomycetes such as *Streptomyces* pSAM2 [46]. These self-transmissible elements rely on a single FtsK/SpoIIIE-like protein channel that translocates double-stranded molecules into adjacent cells within the hyphae of these filamentous bacteria. Hence to prevent loss in the donor cell, they replicate using a dedicated *oriV* and rolling-circle replication initiator proteins, such as RepSA, RepAM or Rep2, that seem to be expressed only after excision from their host chromosome [47,48]. *Staphylococcus aureus* pathogenicity islands (SaPIs) and recently discovered phage-inducible chromosomal islands (PICIs) in *Enterobacteriaceae* and *Pasteurellaceae* are phage satellites that parasite temperate helper bacteriophages for their own dissemination [49]. Like SGI1, SaPIs and PICIs excise from the chromosome of their host and undergo active replication in the presence of their helper element. Their replication, which is usually mediated by a primase and replicase that they encode, is necessary to allow packaging of the phage satellite genome into viral particles. Whereas in all these instances, the biological role of replication is clear, the function ensured by SGI1 replication remains uncertain. The need for a dedicated replicon on SGI1 for the sole purpose of retaining a copy of SGI1 in donor cells after conjugative transfer seems to be overkill as this function is expected to be ensured by the template strand during intracellular rolling-replication initiated at *oriT*, unless SGI1 lacks a functional single-strand origin (*sso*) that could be required to resynthesize the transferred strand from the template strand as shown for Tn*916* or ICE*Bs1* [43,50]. Nevertheless, the discovery of SGI1-like elements with alternative *rep* genes (Fig 6) suggests an important role of replication in the life cycle of genomic islands of the SGI1 family at large.

Unambiguously, replication of SGI1 has a destabilizing effect on IncC plasmid maintenance. In addition, replication functions seem to play a role in the reduction of cotransfer of SGI1 and its helper plasmid, which in return could help stabilize SGI1 in its new host by preventing futile excision and eventual loss. Furthermore, as shown previously, SGI1 reshapes the mating pore encoded by the IncC plasmid to promote its self-propagation. SGI1 encodes alternative TraN, TraG, and TraH subunits, which results in the replacement of the cognate subunits encoded by the IncC plasmid in the mating pore [18]. This substitution was shown to enhance the transmissibility of SGI1. High SG1 copy number could increase the production of $TraN_S$, $TraG_S$, and $TraH_S$, facilitating the alteration of the mating pore to enhance SGI1 propagation at the expense of the helper plasmid. Consistent with this hypothesis, deletion of *rep* or *oriV* suppressed the inhibition of IncC plasmid transfer phenotype (Fig 3F).

SGI1 replication is linked to its incompatibility with IncC plasmids, and one may wonder what evolutionary benefit SGI1 can get from destabilizing its helper element. IncC plasmids mobilize other genomic islands, such as the multidrug resistance island MGI*Vch*Hai6 found in clinical non-O1/non-O139 *V. cholerae* isolates [20,51–53]. Yet, unlike SGI1, MGI*Vch*Hai6 lacks a *rep* homologue and transfers at a much lower rate [51]. Furthermore, MGI*Vch*Hai6 is not as prevalent in Gammaproteobacteria. One reason could be that by allowing the co-residence of its helper plasmid in the cell, MGI*Vch*Hai6 is subjected to a high rate of excision that undermines its stability during cell division. On the contrary, SGI1 prevents long-term co-residence of its helper plasmid, as it has been shown to inhibit co-transfer and promote plasmid loss [26]. A key element of SGI1 epidemiological success could be the association between replication and entry exclusion evasion. SGI1 evades entry exclusion of IncC plasmids, allowing its transfer to recipient cells containing an IncC plasmid [18,19]. Upon entry into such cells,

SGI1 cannot stably integrate into the chromosome. Instead, SGI1 actively replicates which could favor its epidemic spread into neighboring IncC$^+$ cells, thanks to entry exclusion evasion. Ultimately, by triggering population-wide helper plasmid loss, SGI1 prevents *xis* and *rep* expression due to AcaCD depletion, thereby promoting its stabilization through integration into the host chromosome.

SGI1 and its siblings seem to have evolved a fiery love/hate relationship with their helper plasmids, relying exclusively on them to disseminate while preventing their cotransfer and mid- and long-term coexistence within the same cell. Based on our results, SGI1 replication likely plays an important role in this complex interaction. Even so, how exactly replication affects plasmid stability remains to be unraveled. One key aspect to investigate is the influence of copy number shift and how it affects gene expression and production of SGI1-specific proteins that may interfere with the replication, partition in daughter cells and transfer of IncA and IncC plasmids.

## Materials and methods

### Strains, media and antibiotics

The strains used in this study are listed in Table 1. The strains were routinely grown in lysogeny broth (LB-Miller, EMD) at 37˚C in an orbital shaker/incubator and were preserved at -80˚C in LB broth containing 30% (vol/vol) glycerol. Antibiotics were used at the following concentrations: ampicillin (Ap), 100 μg/ml; chloramphenicol (Cm), 20 μg/ml; kanamycin (Kn), 50 μg/ml; nalidixic acid (Nx), 40 μg/ml; rifampicin (Rf), 50 μg/ml; spectinomycin (Sp), 50 μg/ml; tetracycline (Tc), 12 μg/ml. When required, bacterial cultures were supplemented with 0.02% L-arabinose.

### Molecular biology methods

Genomic and plasmid DNA were prepared using the QIAmp DNA Mini Kit (Qiagen) and EZ-10 Spin Column Plasmid DNA Minipreps Kit (Biobasic), respectively, according to the manufacturer's instructions. Oligonucleotide primers were purchased from Integrated DNA Technologies. When necessary, PCR products were purified using the QIAquick PCR Purification Kit (Qiagen) according to the manufacturer's instructions. Ligations were performed using T4 DNA ligase (New England Biolabs) with cloning vectors that were dephosphorylated using Antarctic Phosphatase (New England Biolabs) according to the manufacturer's instructions. *E. coli* was transformed by electroporation according to Dower *et al*. [58]. Electroporation was carried out in a BioRad GenePulser Xcell apparatus set at 25 μF, 200 V and 1.8 kV using 1-mm gap electroporation cuvettes. Sequencing reactions were performed by the Plateforme de Séquençage et de Génotypage du Centre de Recherche du CHUL (Québec, QC, Canada).

### Plasmid constructions

Plasmids and primers used in this study are described in Table 1 and S1 Table, respectively. PCR fragments were amplified using Q5 High-Fidelity DNA Polymerase (New England Bio-Labs). *mCherry* and *mNeonGreen* were respectively amplified using primer pairs Fw-EcoRI-RBSMC/Rv-MC-KpnI and Fw-EcoRI-RBSNG/Rv-NG-KpnI from plasmids pmCherry (Takara) and pMFflT-o4-neonGreen, respectively. These PCR fragments were digested with EcoRI and KpnI, and cloned into pBAD30 digested with the same enzymes, yielding pRed and pGreen with the reporter genes under the control of the $P_{BAD}$ promoter. To construct pRedKnFRT and pGreenKnFRT, pKD4 was digested by HindIII and PvuI to recover the

**Table 1. *E. coli* K-12 derivative strains, plasmids and genomic islands used in this study.**

| Strain, plasmid or genomic island | Relevant genotype or phenotype[a] | Source or reference |
|---|---|---|
| *Strains* | | |
| BW25113 | F⁻ Δ(*araD-araB*)567, Δ*lacZ4787*(::*rrnB-3*), λ⁻, *rph-1*, Δ(*rhaD-rhaB*)568, *hsdR514* | [54] |
| KH95 | BW25113 *rpoB526* (Rf) | This study |
| VB113 | Nx-derivative of BW25113 (Nx) | [55] |
| *Plasmids* | | |
| pKD3 | PCR template for one-step chromosomal gene inactivation (Cm) | [54] |
| pKD4 | PCR template for one-step chromosomal gene inactivation (Kn) | [54] |
| pVI42B | pVI36 BamHI::*P_{lac}*-*lacZ* | [56] |
| pMFflT-o4-neonGreen | PCR template for *mNeonGreen* amplification (Tc) | S. Rodrigue |
| pMS1 | pSC101 *cI857*; *P_L*-*gam-bet-exo*; *gen*; λRed expression vector (Ts, Gm) | [55] |
| pBAD30 | *ori*_{p15A} *araC P_{BAD}* (Ap) | [57] |
| pVCR94 | IncC conjugative plasmid (Su Tm Cm Ap Tc Sm) | [55] |
| pVCR94^Sp | Sp^R derivative of pVCR94 (Sp Su) | [19] |
| pVCR94^GreenSp | pVCR94^Sp *traG_c*Ω(*P_{BAD}*-*mNeonGreen*-FRT) (Sp Su) | This study |
| pVCR94^GreenKn | pVCR94 *traG_c*Ω(*P_{BAD}*-*mNeonGreen*-FRT-*aph*-FRT) (Kn Su Tm Cm Ap Tc Sm) | This study |
| pVCR94^Green | pVCR94 *traG_c*Ω(*P_{BAD}*-*mNeonGreen*-FRT) (Su Tm Cm Ap Tc Sm) | This study |
| pRed | pBAD30::*mCherry* (Ap) | This study |
| pGreen | pBAD30::*mNeonGreen* (Ap) | This study |
| pRedKnFRT | pRed::FRT-*aph*-FRT (Ap Kn) | This study |
| pGreenKnFRT | pGreen::FRT-*aph*-FRT (Ap Kn) | This study |
| p*acaCD* | pBAD30::*acaCD* | [20] |
| p*int* | pBAD30::*int* (*S001*) | This study |
| p*xis* | pBAD30::*xis* (*S002*) | This study |
| p*rep* | pBAD30::*rep* (*S003*) | This study |
| pSGI1 | SGI1^Red Δ*int*::*aph* (Cm Kn) resulting from post-excision *int* deletion | This study |
| *Genomic islands* | | |
| SGI1 | SGI1 inserted at the 3' end of *trmE* (Ap Cm Sp Sm Su Tc) | [3] |
| SGI1^Kn | ΔIn104::*aph* mutant of SGI1 devoid of the integron In104 (Kn) | [18] |
| SGI1^Cm | ΔIn104::*cat* mutant of SGI1 devoid of the integron In104 (Cm) | This study |
| SGI1^RedKn | SGI1^Cm *S009*Ω(*P_{BAD}*-*mCherry*-FRT-*aph*-FRT) (Cm Kn) | This study |
| SGI1^Red | SGI1^Cm *S009*Ω(*P_{BAD}*-*mCherry*-FRT) (Cm) | This study |
| SGI1^Red Δ*int* | Δ*S001*::*aph* mutant of SGI1^Red (Cm Kn) | This study |
| SGI1^Red Δ*xis* | Δ*S002*::*aph* mutant of SGI1^Red (Cm Kn) | This study |
| SGI1^Red Δ*rep* | Δ*S003*::*aph* mutant of SGI1^Red (Cm Kn) | This study |
| SGI1^Red Δ*oriV* | Δ*oriV*::*aph* mutant of SGI1^Red (Cm Kn) | This study |

[a] Ap, ampicillin; Cm, chloramphenicol; Gm, gentamycin; Kn, kanamycin; Nx, nalidixic acid; Rf, rifampicin; Sp, spectinomycin; Sm, streptomycin; Su, sulfamethoxazole; Tc, tetracycline; Tm, trimethoprim; Ts, thermosensitive.

1,645-bp HindIII fragment (Flp recombination target (FRT)-flanked *aph* resistance gene cassette (Kn^R)) that was subsequently cloned into pRed and pGreen cut with HindIII. To

construct the expression vectors p*int*, p*xis* and p*rep*, PCR fragments containing *int*, *xis* or *rep* were amplified from genomic DNA of *E. coli* MG1655 bearing SGI1 as the template and primer pairs SGI1intEcoRI.for/SGI1intEcoRI.rev, SGI1xisEcoRIb.for/SGI1xisEcoRI.rev, and Fw-KpnI-Rep/Rv-Rep-SalI, respectively. The PCR fragments were digested by EcoRI, or KpnI and SalI and cloned into pBAD30 cut with the same enzymes. The integrity of all resulting plasmids was confirmed by restriction profiling and DNA sequencing.

## Construction of insertion and deletion mutants

All insertion and deletion mutants in pVCR94 and SGI1 were constructed in *E. coli* BW25113 using the one-step chromosomal gene inactivation technique [54]. The λRed recombination system was expressed using pMS1 as described by Datta *et al.* [59]. All mutations were designed to be non-polar. SGI1$^{Cm}$, a Cm-resistant derivative of SGI1 devoid of In104 was obtained using primer pair SGI1In104cm2.f/SGI1In104cm2.r and pKD3 as the template. SGI1$^{RedKn}$ was constructed by introducing *araC-P$_{BAD}$::mCherry*-FRT-*aph*-FRT into the intergenic region between *S009* and *S010* in SGI1$^{Cm}$ using primer pair FwpBADInsMCSGI1/RvpBADInsMCSGI1 and pRedKnFRT as the template. pVCR94$^{GreenKn}$ and pVCR94$^{GreenSp}$ were constructed by introducing *araC-P$_{BAD}$::mNeonGreen*-FRT-*aph*-FRT between *traG* and *eexC* in pVCR94 and pVCR94$^{Sp}$ using primer pair FwpBADInsNGpVCR/RvpBADInsNGpVCR, and pGreenKnFRT as the template. The FRT-flanked Kn$^R$ cassette was removed by Flp-catalyzed excision using pCP20 [54] to generate SGI1$^{Red}$ and pVCR94$^{Green}$. Deletion of *int*, *xis*, *rep*, *traN$_S$* and *oriV* in SGI1$^{Red}$ was carried out using the same technique with primer pairs SGI1delint.for/SGI1delint.rev, SGI1delxis.for/ SGI1delxis.rev, SGI1deIRep.for/SGI1deIRep.rev, SGI1dels005.for/SGI1dels005.rev, and Fw-DelOriRSGI1/Rv-DelOriRSGI1, respectively, and pKD4 as the template. The translational fusions Δ(*xis-oriV*)-*rep*'-'*lacZ* and *rep*'-'*lacZ* in SGI1$^{Kn}$ were constructed using primer pairs rep-lacZ.f/rep-lacZdelxis.r2 and rep-lacZ.f/rep-lacZ.r2, and pVI42B as the template. Scars in all constructions were determined by PCR and Sanger sequencing.

## Cohabitation assays

All cohabitation experiments were done as follows. The strains containing both elements were inoculated in LB broth with selective pressure (Kn for SGI1$^{RedKn}$ or SGI1$^{Red}$ deletion mutants, and Tc for pVCR94$^{Green}$) and arabinose for induction of the reporter genes, and grown overnight at 37˚C. On the next day, the cultures were used to inoculate (1:2,000 dilution) fresh LB broth with arabinose without antibiotics and incubated at 37˚C. Furthermore, the overnight grown culture (stationary phase cells) was diluted 1:1,000 in PBS and analyzed by flow cytometry. This quantification was defined as the initial population composition (G0). This population was considered as pure and suitable for the experiment if more than 95% of the cells were positive for green and red fluorescence. Subsequently, cultures were passaged twice a day in fresh medium, which equals approximately 9 generations per passage. Flow cytometry analyses were done on stationary phase cells at days 1, 2 and 3, corresponding approximately to generations 18, 36 and 54. At the end of the experiment, the last passage was performed with selective pressure to restore the initial population. This control confirmed that loss of fluorescence resulted from element instability, not from mutations in the reporter genes.

For mutant complementation assays, the same conditions were used except that pVCR94$^{Green}$ was replaced with pVCR94$^{GreenSp}$ due the Ap$^R$ phenotype conferred by both pVCR94$^{Green}$ and pBAD30.

## Flow cytometry

Culture samples were diluted 1:1,000 in 500 μl of PBS. Fluorescence intensity of NeonGreen and mCherry in cells was monitored by flow cytometry analysis on a BD FACSJazz (BD Biosciences), and data were acquired with the BD FACS Sortware. mNeonGreen and mCherry were excited with 488 and 561 nm solid-state lasers, and their emission was detected using 513/17 and 610/20 nm emission filters, respectively. For each sample, fluorescence of 20,000 cells was captured, and the data was analyzed using FCS Express 7 (De Novo Software).

## RNA extraction and cDNA synthesis

Briefly, RNA extractions were performed as follows. *E. coli* KH95 containing SGI1$^{Red}$ with or without p*acaCD* was grown at 37˚C for 16 h in LB broth containing the appropriate antibiotics. The cultures were diluted 1:100 in fresh medium containing the appropriate antibiotics and grown to an $OD_{600}$ of 0.2 before being diluted 1:10 again in fresh medium containing the appropriate antibiotics and supplemented with 0.02% arabinose when needed. After a 2h incubation period, 1 ml of the culture was used for total RNA extraction using Direct-zol RNA extraction kit (Zymo Research) and TRI Reagent (Sigma-Aldrich) according to the manufacturer's instructions. Once purified, the RNA samples were treated using 2 units of DNase I (New England Biolabs) according to the manufacturer's instructions to eliminate any residual gDNA. cDNA was synthesized from 0.2 μg of RNA and 2 pmol of gene-specific primer rep_RT (Integrated DNA Technologies), using the reverse transcriptase SuperScript III (Invitrogen), according to the manufacturer's instructions. Control reactions without reverse transcriptase treatment ('noRT') were performed for each sample. PCR reactions aiming at amplifying *rep*, *S004* and *traN$_S$* were carried out using cDNAs as described in Garriss *et al*. [60].

## β-galactosidase assays

Qualitative assays were performed by depositing 10μl aliquots of overnight cultures with appropriate antibiotics on solid agar supplemented with 5-bromo-4-chloro-3-indolyl-β-D-galactopyranoside (X-gal) with or without 0.02% arabinose. The plates were observed after an overnight incubation at 37˚C. Quantitative assays were performed with 2-nitrophenyl-β-D-galactopyranoside (ONPG) according to a protocol adapted from Miller [61]. After an overnight incubation at 37˚C with appropriate antibiotics with or without 0.2% arabinose, cultures were diluted 1:100 in 50ml LB broth with appropriate antibiotics with or without 0.2% arabinose and grown until an $OD_{600}$ of 0.2 was reached. Two series of 1:10 dilutions were then prepared in total volumes of 5ml LB broth supplemented with appropriate antibiotics with or without 0.2% arabinose and incubated for 2 hours at 37˚C. Values are expressed as the mean and standard error of the mean values calculated from three biological replicates.

## qPCR

Quantification of the copy number of pVCR94 as well as excision rate and copy number of SGI1 forms (circular or integrated) were assessed by real-time quantitative qPCR. Primers used in this quantification are listed in S1 Table. The data were analyzed using the threshold cycle ($\Delta\Delta C_T$) method using the three target genes *dnaB*, *hicB* and *trmE* as chromosomal references. Copy number of pVCR94 and SGI1 was assessed using *repA* and *sgiA* as the targets, respectively. Excision of SGI1 was assessed using the chromosomal free *attB* site as the target. qPCR experiments were performed in triplicate on the RNomics Platform of the Laboratoire

de génomique fonctionnelle de l'Université de Sherbrooke (https://rnomics.med.usherbrooke.ca) (Sherbrooke, QC, Canada).

## Identification of *oriV*

Search for direct and inverted sequence repeats in the region downstream of *S003* was carried out using the genomic similarity search tool Yass [62] available at https://bioinfo.lifl.fr/yass/index.php and the MEME motif discovery algorithm [30] available at http://meme-suite.org/index.html. Identification of putative DnaA boxes was carried out using MAST [63] with the DnaA motif matrix (accession MX000098) obtained from PRODORIC Release 8.9. Prediction of Rho-independent transcription terminator was carried out using ARNold [64] available at http://rssf.i2bc.paris-saclay.fr/toolbox/arnold/index.php.

## Supporting information

**S1 Fig. Impact of *int* on SGI1 replication.** (A) Evolution of the percentage of *E. coli* KH95 cells bearing pVCR94$^{Green}$ and SGI1$^{Red}$ $\Delta int$ over 54 generations in the absence of antibiotics as monitored using FC. (B) Complementation of SGI1$^{Red}$ $\Delta int$ with p*int*. KH95 carried pVCR94$^{GreenSp}$ in these assays.
(TIF)

**S2 Fig. Impact of *traN$_S$* on SGI1 replication.** Evolution of the percentage of *E. coli* KH95 cells bearing SGI1$^{Red}$ $\Delta traN_S$ and pVCR94$^{Green}$ over 54 generations in the absence of antibiotics as monitored using FC.
(TIF)

**S3 Fig. Impact of *xis* on SGI1 replication.** (A) Evolution of the percentage of *E. coli* KH95 cells bearing pVCR94$^{Green}$ and SGI1$^{Red}$ $\Delta xis$ over 54 generations in the absence of antibiotics as monitored using FC. (B) Complementation of SGI1$^{Red}$ $\Delta xis$ with p*xis*. KH95 carried pVCR94$^{GreenSp}$ in these assays.
(TIF)

**S4 Fig. Impact of *rep* on SGI1 replication.** (A) Evolution of the percentage of *E. coli* KH95 cells bearing pVCR94$^{Green}$ and SGI1$^{Red}$ $\Delta rep$ over 54 generations in the absence of antibiotics as monitored using FC. (B) Complementation of SGI1$^{Red}$ $\Delta rep$ with p*rep*. KH95 carried pVCR94$^{GreenSp}$ in these assays.
(TIF)

**S5 Fig. Evolution of the percentage of *E. coli* KH95 cells bearing pSGI1 and pVCR94$^{GreenSp}$ over 54 generations in the absence of antibiotics as monitored using FC.**
(TIF)

**S6 Fig. *rep* expression is controlled by AcaCD.** (A) Sequence of the *rep*'-'*lacZ* translational fusion. The open reading frames are indicated by arrows. The predicted Shine-Dalgarno sequence of *rep* is underlined and its start codon is shown in bold. The red asterisk indicates the stop codon (in red) of *S004*. The sequence of *lacZ* is shown in blue whereas the sequence of SGI1 is shown in black. Predicted translation product are shown below the nucleotide sequence. (B) Schematic representation of the *rep*'-'*lacZ* translational fusion in SGI1. (C) β-galactosidase assays of the translational *rep*'-'*lacZ* fusion in SGI1$^{Kn}$ performed in IncC-free cells (-), and in the presence of pVCR94$^{Sp}$, its $\Delta acaCD$ mutant or p*acaCD* without or with arabinose (+ara).
(TIF)

**S7 Fig. Impact of *oriV* on SGI1 replication.** Evolution of the percentage of *E. coli* KH95 cells bearing pVCR94$^{Green}$ and SGI1$^{Red}$ Δ*oriV* over 54 generations in the absence of antibiotics as monitored using FC.
(TIF)

**S1 Table Primers used in this study.**
(DOCX)

## Acknowledgments

We are grateful to Sébastien Rodrigue for the kind gift of pMFflT-o4-neonGreen. We acknowledge all lab members for thoughtful discussions. We also thank Malika Humbert for her technical assistance and Alain Lavigueur for his insightful comments on the manuscript.

## Author Contributions

**Conceptualization:** Kévin T. Huguet, Vincent Burrus.

**Data curation:** Kévin T. Huguet.

**Formal analysis:** Kévin T. Huguet, Nicolas Rivard, Jason Palanee, Vincent Burrus.

**Funding acquisition:** Vincent Burrus.

**Investigation:** Kévin T. Huguet, Nicolas Rivard, Jason Palanee, Vincent Burrus.

**Methodology:** Kévin T. Huguet, Nicolas Rivard, Daniel Garneau, Vincent Burrus.

**Project administration:** Vincent Burrus.

**Resources:** Vincent Burrus.

**Supervision:** Vincent Burrus.

**Validation:** Kévin T. Huguet, Nicolas Rivard, Daniel Garneau, Vincent Burrus.

**Visualization:** Daniel Garneau, Vincent Burrus.

**Writing – original draft:** Kévin T. Huguet.

**Writing – review & editing:** Kévin T. Huguet, Nicolas Rivard, Daniel Garneau, Vincent Burrus.

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
