## [Decision Letter · Decision Letter 0]

23 Jun 2020

Dear Dr Burrus,

Thank you very much for submitting your Research Article entitled 'Replication of the Salmonella Genomic Island 1 (SGI1) triggered by helper IncC conjugative plasmids promotes incompatibility and plasmid loss' to PLOS Genetics. Your manuscript was fully evaluated at the editorial level and by independent peer reviewers. The reviewers appreciated the attention to an important topic but identified some aspects of the manuscript that should be improved.

We therefore ask you to modify the manuscript according to the review recommendations before we can consider your manuscript for acceptance. Your revisions should address the specific points made by each reviewer.

[LINK]

Yours sincerely,

Diarmaid Hughes

Associate Editor

PLOS Genetics

Lotte Søgaard-Andersen

Section Editor: Prokaryotic Genetics

PLOS Genetics

Reviewer's Responses to Questions

**Comments to the Authors:**

Reviewer #1: Huguet and coworkers study the apparent incompatibilities between the Salmonella genomic island SIG1 and plasmids of the type IncC/IncA. They label SIG1 and plasmid with different inducible fluorescent reporter proteins to quantify presence of both elements in single cells by flow cytometry under different conditions and various mutant backgrounds. This was further backed up by qPCR and other DNA analyses, and finally by proposing similarities to related systems.

The main conclusion of the work is that SIG1 is replicating in its excised state, which is dependent on presence of an IncC-type plasmid with a transcription activator that controls expression of the SIG1 rep-gene. Yet, this replication causes an incompatibility, which leads to preferential loss of the plasmid.

COMMENTS

1) I have very few criticisms on this work. It is very complete and clearly written. The results are convincing. There are clear differences in the expression levels of the fluorescence markers, which allows the authors to conclude that must correspond to three SIG1 states: a chromosomally integrated state (low fluo); an excised state (intermediate fluo) and a replicated excised state (high fluorescence). The plasmid leads to two fluorescent states: presence (high) or absence (low). This was benchmarked by qPCRs, by isolation of the physical SIG1 form, by different mutations that block the SIG1 from excision or from replicating.

The causal link of SIG1 replication after excision to induction of the Rep factor on SIG1 by the IncC plasmid AcaCD activator complex is very clearly demonstrated as well.

2) The results of this work therefore clearly advance our understanding of integrated mobilizable elements that rely on other DNA replicons for their transfer. It is extremely curious that an element like SIG1 needs a plasmid for its excision and for its replication, and then hijacks the conjugative system of the plasmid by replacing part by its own conjugation proteins such that its own transfer is preferred over that by the plasmid. Parasites within parasites!

Apart from the molecular intricacies, the fate of SIG1-type elements is very relevant for spread of antimicrobial compound resistance factors, which are often encoded on them.

3) Having said this: there must be some downsides to the SIG1-type strategy. What if there is cross-activation by a chromosomally integrated acaCD-type gene? Would that lead to loss of SIG1 or would it continue to replicate extra-chromosomally? Did the authors ever try such an experiment?

The flow cytometry data seem to suggest that there is some escape in the 'incompatibility' conquest between excised replicating SIG1 and IncC. What happens in cells where IncC continues to be present? Are there any mutations inactivating SIG1 or is it simply a matter of chance?

4) l. 120 Why are replication-deficient mutants not evolving spontaneously? You mention here and before that no wild-type strains have been found that carry SIG1 and an IncA/C plasmid.

5) l. 142-144 maybe this is trivial, but how can you select for the presence of both elements that are incompatible in their replication? Will this not automatically lead to appearance of compensatory mutations somewhere?

6) l. 230 Bit an awkward notation to write 8.38 x 10-5%. Maybe write both this and 81% in l. 231 as a proportion.

7) l. 241/242: is there no interference at all by inducing all these different elements by arabinose?

8) l. 472. Maybe not so uncertain. The big question seems to be what the actual molecular mechanism is that causes SIG1 replication to interfere with that of IncC. It seems to be a very gradual mechanism (i.e., taking more than 50 generations to lead to most of the reduction in a population), which might be due to the continued reliance of SIG1 on IncC? (If a cell would no longer have IncC, the AcaCD would no longer be expressed and SIG1 is out of business). What happens after SIG1 transfer? Is it lost from a 'donor' cell?

9) The figures included in the produced PDF are of embarrasing low quality. However, this happened to us recently as well (which was held against us), but we noticed that this is a mistake in the PLoS Genetics conversion process and should absolutely be brought to the attention of the editorial team and improved.

10) Fig. 1. I noticed in this figure and in some others (e.g., Fig. 2C bottom right) that the FSC signal of the mChe-cells is sometimes lower than that of mNeon (panel C). In which growth phase are these cells measured? Do they have a slightly different cell size?

11) Fig. 3D: why are these effects not tested in an ANOVA that takes all mutants and wild-type into consideration, rather than pair-wise t-tests? That would be better practice.

Reviewer #2: This is a very interesting paper analysing the dynamics of two different types of MGEs, the Salmonella genomic island 1 (SGI1) and the IncC plasmids. Interestingly, while SGI1 requires IncC plasmids for mobility, these two elements are incompatible. This manuscript analyses the genetic causes of this incompatibility. Overall, this is a nice piece of work, and I just have some minor comments that hopefully will improve the manuscript:

- In a very low percentage of cells, both elements may persist, suggesting they have mutated to avoid the incompatibility. It would be nice to sequence some of these stable elements. This could identify additional genes involved in the process.

- Are the authors completely convinced that the SGI int mutant does not replicate in situ? Maybe this could be specifically tested, using the qPCR utilized in the study for the wt plasmid, for example.

- Is Rep (together with its cognate oriV site) sufficient for autonomous SGI1 replication? Maybe the authors could clone these two elements in a suicide plasmid to test if they can support SGI1 replication.

- I would like to see some discussion about the evolutionary advantage of this relationship, especially for the island? It is nice but strange that one parasitic element, the island, which requires the plasmid for transfer, can’t cohabit with the helper plasmid.

Reviewer #3: These authors present a careful and clever analysis of co-existence of conjugative plasmids and ICE-like elements in Salmonella. Overall, I thought the paper makes a nice contribution to the field and is well written (albeit maybe a little wordy), with nice illustrations and figure design. I have no major problems with the ms as presented here, and my comments refer to potential follow up studies that might give a little more impact to this paper. They are as follows:

1) In the primary experiment shown in Fig. 2, the authors were able to select out rare clones (undetectable by flow cytometry) where both elements were still co-resident-- it would be interesting to plate these on antibiotic plates, select some colonies (maybe 10 or so) and repeat the passage in the absence of selection to see if they get the same result-- it is possible that there could have been selections for suppressor mutants of the replication/incompatibility functions encoded by the wild type elements and these might further inform the mechanisms.

2) The authors might have published this previously, but it would be interesting t know whether there is a basal level of transfer (eg. solid surface matings) of the Island in the absence of a helper plasmid..

Otherwise I thought this was a nice paper.

**Have all data underlying the figures and results presented in the manuscript been provided?**

Reviewer #1: Yes

Reviewer #2: Yes

Reviewer #3: Yes

PLOS authors have the option to publish the peer review history of their article (what does this mean?). If published, this will include your full peer review and any attached files.

Reviewer #1: No

Reviewer #2: No

Reviewer #3: No

---

## [Editor Report · Decision Letter 1]

30 Jun 2020

Dear Dr Burrus,

We are pleased to inform you that your manuscript entitled "Replication of the Salmonella Genomic Island 1 (SGI1) triggered by helper IncC conjugative plasmids promotes incompatibility and plasmid loss" has been editorially accepted for publication in PLOS Genetics. Congratulations!

Yours sincerely,

Diarmaid Hughes

Associate Editor

PLOS Genetics

Lotte Søgaard-Andersen

Section Editor: Prokaryotic Genetics

PLOS Genetics

Comments from the reviewers (if applicable):

**Data Deposition**

http://datadryad.org/submit?journalID=pgenetics&manu=PGENETICS-D-20-00839R1

**Press Queries**

---

## [Editor Report · Acceptance letter]

21 Jul 2020

PGENETICS-D-20-00839R1 

Replication of the Salmonella Genomic Island 1 (SGI1) triggered by helper IncC conjugative plasmids promotes incompatibility and plasmid loss 

Dear Dr Burrus, 

We are pleased to inform you that your manuscript entitled "Replication of the Salmonella Genomic Island 1 (SGI1) triggered by helper IncC conjugative plasmids promotes incompatibility and plasmid loss" has been formally accepted for publication in PLOS Genetics! Your manuscript is now with our production department and you will be notified of the publication date in due course.

With kind regards,

Jason Norris

PLOS Genetics

On behalf of:
